# Case-control study of patient characteristics, knowledge of the COVID-19 disease, risk behaviour and mental state in patients visiting an emergency room with COVID-19 symptoms in the Netherlands

J. P. M. van der Valk[1]*, F. W. J. Heijboer[1], H. van Middendorp[2], A. W. M. Evers[2], J. C. C. M. in 't Veen[1]

1 Department of Pulmonary Medicine, Franciscus Gasthuis & Vlietland, Rotterdam, The Netherlands,
2 Department of Health, Medical and Neuropsychology, Leiden University, Leiden, The Netherlands

* h.kuiper-vandervalk@franciscus.nl

## Abstract

### Background

Coronavirus disease 2019 is a serious respiratory virus pandemic. Patient characteristics, knowledge of the COVID-19 disease, risk behaviour and mental state will differ between individuals. The primary aim of this study was to investigate these variables in patients visiting an emergency department in the Netherlands during the COVID-19 pandemic and to compare the "COVID-19 suspected" (positive and negative tested group) with the "COVID-19 not suspected" (control group) and to compare in the "COVID-19 suspected" group, the positive and negative tested patients.

### Methods

Consecutive adult patients, visiting the emergency room at the Franciscus Gasthuis & Vlietland, Rotterdam, the Netherlands, were asked to fill out questionnaires on the abovementioned items on an iPad. The patients were either "COVID-19 suspected" (positive and negative tested group) or "COVID-19 not suspected" (control group).

### Results

This study included a total of 159 patients, 33 (21%) tested positive, 85 (53%) negative and 41 (26%) were COVID-19 not suspected (control group). All patients in this study were generally aware of transmission risks and virulence and adhered to the non-pharmaceutical interventions. Working as a health care professional was correlated to a higher risk of SARS-Cov-2 infection (p- value 0.04). COVID-19 suspected patients had a significantly higher level of anxiety compared to COVID-19 not suspected patients (p-value < 0.001). The higher the anxiety, the more seriously hygiene measures were followed. The anxiety

**Data Availability Statement:** All relevant data are within the manuscript and its Supporting information files.

**Funding:** There was not funding for this study.

**Competing interests:** There are no competing interests.

**Abbreviations:** CI, Confidence interval; COVID-19, Coronavirus disease 2019; ICU, Intensive care unit; NPI, Non-pharmaceutical interventions; OR, Odds Ratio; PCR, Polymerase chain reaction.

scores of the patients with (pulmonary) comorbidities were significantly higher than without comorbidities.

## Conclusion

This is one of the first (large) study that investigates and compares patient characteristics, knowledge, behaviour, illness perception, and mental state with respect to COVID-19 of patients visiting the emergency room, subdivided as being suspected of having COVID-19 (positive or negative tested) and a control group not suspected of having COVID-19. All patients in this study were generally aware of transmission risks and virulence and adhered to the non-pharmaceutical interventions. COVID-19 suspected patients and patients with (pulmonary) comorbidities were significantly more anxious. However, there is no mass hysteria regarding COVID-19. The higher the degree of fear, the more carefully hygiene measures were observed. Knowledge about the coping of the population during the COVID-19 pandemic is very important, certainly also in the perspective of a possible second outbreak of COVID-19.

## Introduction

Coronavirus disease 2019 (COVID-19) is caused by the SARS-Cov-2 virus and constitutes the most serious respiratory virus pandemic since the 1918 H1N1 influenza pandemic. COVID-19 is characterized by fever and respiratory symptoms like cough, sneeze, and shortness of breath [1]. SARS-Cov-2 virus spreads by transmission of respiratory (small) droplets containing the virus particles from person to person, mostly in close contact [2]. Symptomatic patients spread the virus very fast. The SARS-Cov-2 virus can also be transmitted by asymptomatic persons and by contact with contaminated surfaces. Occasionally, the virus can be transmitted from humans to animals and vice versa [3]. Prevention of transmission is very important in the absence of an effective COVID-19 vaccine or treatment.

Non-pharmaceutical interventions (NPIs) to avoid transmission of SARS-Cov-2 involve hygiene measures, limitation of human contact, and social distancing. The Dutch National Institute for Public Health and Environment (RIVM) recommends washing hands thoroughly and frequently, coughing and sneezing in the inside of the elbow, avoiding shaking hands, keeping a distance of 1.5 meters (2 arms' lengths) from others, and staying at home as much as possible.

Knowledge of contamination risks, severity of SARS-Cov-2 infection, the chance of recovery and long-term consequences of the disease will differ between individuals, including those individuals with and without a SARS-Cov-2 infection, and might influence the (risk) behaviour of people. Also, the mental state and comorbidities of people may influence (risk) behaviour and consequently the chance of SARS-Cov-2 contamination. Knowledge about the coping of the population during the COVID-19 pandemic is very important, certainly also in the perspective of a possible second outbreak of COVID-19.

The present report concerns an observational study on patient characteristics and knowledge of the disease, risk behaviour, illness perception, and mental state (at the time of assessment) of patients visiting an emergency room in the Netherlands, subdivided as "COVID-19 suspected" (positive and negative tested group) or "COVID-19 not suspected" (control group).

These variables were measured for three groups: those tested positive for SARS-Cov-2, those tested negative, and the control group. The degree of anxiety in the total group has also been correlated to comorbidities and the hygiene measures, human contact, and social distancing.

## Material and method

### Study design and patient selection

This study was designed as an observational questionnaire study and registered in the Dutch Trial Register as **PA**tient's knowledge and behaviour o**N** the COVID-19 disease and as **D**eter-**MI**nants of **C**ontamination (PANDEMIC) study (trial number NL8563). The study was submitted to the medical ethical review board and was considered not subject to the Medical Research Involving Human Subjects Act (WMO; W20.075). We adhered to the methods and procedures of the Strengthening the Reporting of Observational Studies in Epidemiology (STOBE) guidelines for reporting this study. It was not possible to involve patients or the public in the design, or conduct, or reporting, or dissemination plans of our research, because of the very fast procedure in the COVID-19 crisis.

Consecutive adult patients (>18 years of age) visiting the emergency room at the Franciscus Gasthuis & Vlietland, Rotterdam, the Netherlands, were asked to participate in this study (n> 200). The patients were "COVID-19 suspected" or "COVID-19 not suspected" (control group). All patients with upper- or lower respiratory symptoms were considered as possible COVID-19 positive. Before completing the survey, participants gave their written informed consent. Reasons for non-participation were patients with a language barrier or patients who could not operate an iPad or who had a severe medical condition (direct transfer to intensive care) or lack of iPads due to the large number of patients arriving at the emergency room at the same time.

A total of 164 patients were included in the period between April 10th and June 20th, 2020. Patients were asked to fill out their general patient characteristics and answer questions on their knowledge of the disease, risk behaviour, illness perception, and mental status on an iPad with disposable cover to respect hygiene measures. Of these 164 patients, 5 patients did not complete the questionnaires because of logistic–or personal reasons (Fig 1). General patient characteristics and information consisted of year of birth, sex, the number of people in their household, and profession type (if applicable). Vital professions were defined as those in sectors necessary to keeping society running during the COVID-19 crisis, for example: health care, policing, food, waste, and transport. Comorbidities were derived from the electronic patient files.

Patients' knowledge of SARS-Cov-2 was investigated with a self-developed questionnaire on the severity of the disease, route of contamination, and the importance of government measures. Illness perception was measured with questions on the patient's perception of COVID-19 with respect to their specific situation. For example: '*If you are infected with SARS-Cov-2, do you think that you could die from this infection*?' (Attachment 1).

A second questionnaire concerned risk behaviour in terms of hygiene, human contact, and social distancing. The questionnaire was designed to study whether patients adhered to the measures imposed by the Dutch National Institute for Public Health and Environment. For example: *Do you wash your hands frequently; do you avoid group gatherings; do you maintain a 1.5-meter distance between yourself and others*? (Attachment 2).

The last 6-item questionnaire was the validated State-Trait Anxiety Inventory–short (STAI-s) survey [4] to measure the mental state of the COVID-19 suspected patients from the

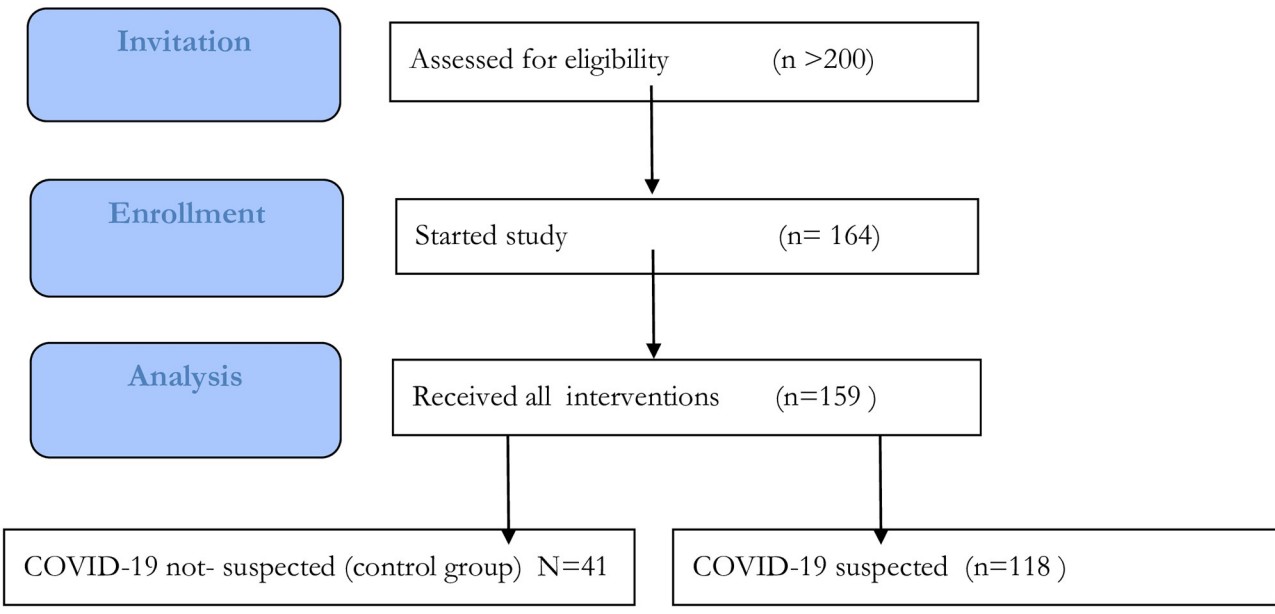

**Fig 1. Flow chart of patient inclusion.**

beginning of the COVID-19 crisis (1st of March 2020) to the time of administering the questionnaires.

The questionnaires were filled out before test results (COVID-19 negative/positive) were known. Three patients did not complete the second questionnaire and 5 patients did not complete the third questionnaire. These data were not part of the analysis.

Patients with a positive nasal Polymerase chain reaction (PCR) swab to SARS-Cov-2 or having SARS-Cov-2 antibodies were considered to be COVID-19 positive.

A comparison is made between the "COVID-19 suspected" (positive and negative tested group) with the "COVID-19 not suspected" (control group) and in the "COVID-19 suspected" group, the positive and negative tested patients.

## Statistical analysis

Patient characteristics were reported in terms of mean, ranges, and proportions. Knowledge, risk behaviour, and illness perception of COVID-19 were reported descriptively. The differences in terms of these factors were compared between 3 groups (those tested positive for SARS-Cov-2, those tested negative, and the control group) using the Kruskal-Wallis test, and were described with p-values. P-values $< 0.05$ were considered as significant.

Mental state was measured by the STAI-s score questionnaire. The sum of the 6 questions was calculated. This score was multiplied by 20/6 to be compared with the long version of the STAI. The One-Way ANOVA test was used to compare the mean scores on anxiety of the 3 groups. A cut off point of 39–40 has been suggested to detect clinically significant anxiety symptoms.

The STAI-s scores were correlated with the risk behaviour score for the areas of hygiene, human contact, and social distancing using linear logistic regression analysis. The risk behaviour score was the sum of the questions divided by the number of questions. The score on a 5-point Likert scale was: 0 points = no risk to 4 points = most risk and on a 2-point Likert scale: 1 point = no risk and 3 points = most risk). The outcomes of the regression analysis were

reported with the regression standardized coefficient (ß), and the CI and p-values. Also, the mean STAI-s score was related to comorbidities with linear logistic regression analysis. Statistical analyses were done using SPSS statistics 24.

# Results

## Study population

This study included a total of 159 patients including 118 COVID-19 suspected patients and 41 COVID-19 not suspected patients (control group). The mean age of all the patients (inclusive control group) was 50.42 years (range 19–86 years), with 70 males (44%) and 89 females (56%) (Table 1). The patients were referred to the Franciscus Gasthuis and Vlietland, Rotterdam, by their general practitioner or taken to hospital directly by ambulance. Of the COVID-19 suspected patients in this study, 33 (21%) patients tested positive and 85 (53%) tested negative for COVID-19.

The average household was 2.76 people (range 1–7) in the total group and was similar for the positive and the negative group. Patient characteristics are summarized in Table 1. Of the SARS-Cov-2 positive cases, 16 patients (48%) had a vital profession (and this differed significantly (p-value 0.04) compared to 23 (26%) in the negative tested group (Fig 2A). More than 60% of these patients worked in the medical sector, mostly in nursing homes or care institutions (56%) (Fig 2B, Table 1). Some of these patients worked with COVID-19 patients and a substantial part worked with colleagues also infected with Sars-Cov2.

## Knowledge/Illness perception

COVID-19 was seen as a severe disease: almost 90% of the total study population believe the disease is worse than influenza. Ninety-eight percent of patients knew that contaminations most frequently occur through person to person contact. Approximately 18% of the patients considered wearing face masks useful in the prevention of SARS-Cov-2 infection. Ninety-four percent of patients thought that adhering to NPIs helps to prevent infection. There was no difference between the positive-tested, negative-tested, and control group with respect to knowledge about COVID-19 (Table 2).

In this study population (COVID-19 suspected and control group), 32% believed they would not be infected by SARS-Cov-2, 37% did not expect to become seriously ill and 42% did

**Table 1. Characteristics of the SARS-Cov-2 positive-tested, negative-tested, and control groups.**

|  | Positive-tested (N = 33) | | Negative-tested (N = 85) | | Control group (N = 41) | |
|---|---|---|---|---|---|---|
| **Age** (Mean, range) | 50.42 | (25–86) | 51.53 | (21–85) | 48.07 | (19–77) |
| **Males** (N, %) | 13 | (39) | 34 | (40) | 23 | (56) |
| **Household** (N, %) | | | | | | |
| 1–2 | 15 | (46) | 50 | (59) | 16 | (39) |
| 3–4 | 14 | (42) | 25 | (29) | 19 | (46) |
| ≥4 | 4 | (12) | 10 | (12) | 6 | (15) |
| **Vital profession** (N, %) | | | | | | |
| Yes | 16 | (48) | 21 | (25) | 11 | (27) |
| **Institution** (N, %) | N = 16 | | N = 21 | | N = 11 | |
| General practice | 1 | (6) | 3 | (14) | 0 | (0) |
| Nursing home /care institution | 9 | (56) | 5 | (24) | 0 | (0) |
| Hospital | 2 | (13) | 3 | (14) | 1 | (9) |
| Not in the medical care sector | 4 | (25) | 10 | (48) | 10 | (91) |

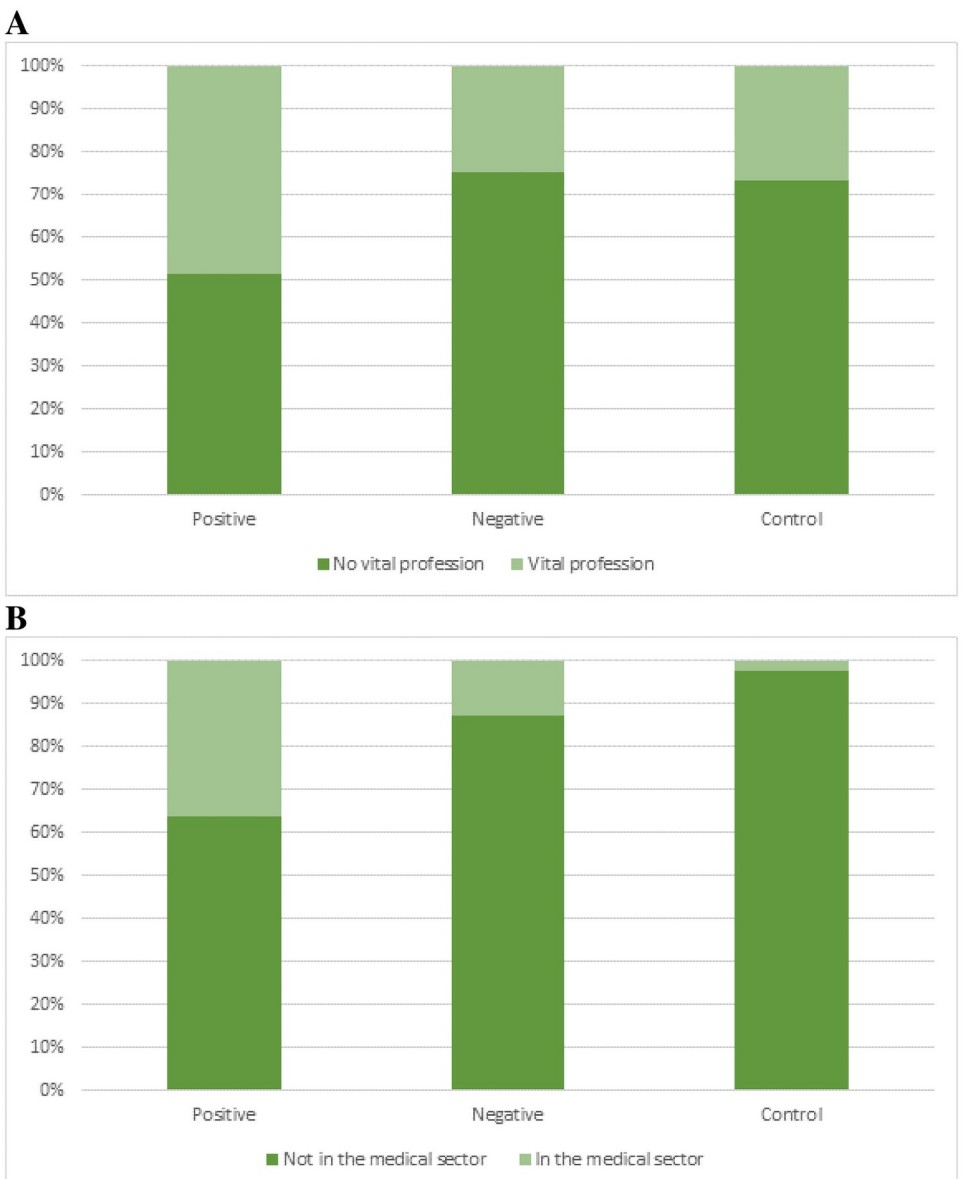

**Fig 2. Patients working in a vital profession (A) and in the medical sector (B) in the SARS-Cov-2 positive-tested, negative-tested, and control groups.** Significantly more patients infected with SARS-Cov-2 work in a vital profession compared to uninfected patients and the control group (p-value 0.04), and most of them work in the medical sector.

not expect to die from the infection. Of all the patients, 20% did not consider the virus to be contagious. The illness perception scores did not differ between the 3 tested groups.

## Risk behaviour

For all patients (COVID-19 suspected- and control group), 43% washed their hands more than 10 times a day and approximately 81% reported washing their hands always/almost always for more than 20 seconds. More than 90% of patients reported using the inside of their elbow to cough and sneeze. Only 11% of patients reported wearing a face mask for protection. Almost all patients indicated always/almost always keeping a 1.5-meter distance from others

**Table 2. Group comparison for non-pharmaceutical interventions (NPIs) and SARS-Cov-2 infection in the positive-tested, negative-tested, and control groups.**

| Behaviour during COVID-19 crisis | N (%) | | N (%) | | N (%) | | P-value |
|---|---|---|---|---|---|---|---|
| | Negative-tested | | Positive-tested | | Control | | |
| **Hygiene** | 85 | | 33 | | 41 | | |
| Washing hands (times a day) | | | | | | | 0.11 |
| 0 | 0 | (0) | 1 | (3) | 1 | (2) | |
| 1–3 | 6 | (7) | 2 | (6) | 3 | (7) | |
| 4–6 | 24 | (28) | 4 | (12) | 16 | (39) | |
| 7–9 | 20 | (24) | 7 | (21) | 6 | (15) | |
| >10 | 35 | (41) | 19 | (58) | 15 | (37) | |
| *Missing* | | | | | | | |
| Washing hands (20 seconds) | | | | | | | 0.59 |
| Never—sometimes | 15 | (18) | 5 | (15) | 10 | (24) | |
| Almost always- always | 70 | (82) | 28 | (85) | 31 | (76) | |
| Stick to coughing rules | | | | | | | 0.14 |
| No-probably | 6 | (2) | 5 | (15) | 4 | (10) | |
| Probably yes- yes | 79 | (98) | 28 | (85) | 37 | (90) | |
| Wearing face mask | | | | | | | 0.42 |
| Never—sometimes | 75 | (88) | 29 | (88) | 37 | (90) | |
| Almost always- always | 10 | (12) | 4 | (12) | 4 | (10) | |
| Keeping 1.5-meter distance | | | | | | | **0.03** |
| Never—sometimes | 2 | (4) | 2 | (6) | 4 | (10) | |
| Almost always- always | 82 | (96) | 31 | (94) | 37 | (90) | |
| *Missing* | 1 | | | | | | |
| **Contacts** | | | | | | | |
| Staying at home | | | | | | | **< 0.001** |
| Never—sometimes | 16 | (19) | 14 | (42) | 13 | (73) | |
| Almost always- always | 67 | (81) | 19 | (58) | 28 | (27) | |
| *Missing* | 2 | | | | | | |
| Shop visits | | | | | | | 0.21 |
| No | 63 | (72) | 26 | (79) | 24 | (60) | |
| Yes | 25 | (28) | 7 | (21) | 16 | (40) | |
| *Missing* | | | | | 1 | | |
| Park/market/ beach visits | | | | | | | **0.07** |
| No | 67 | (81) | 32 | (97) | 35 | (88) | |
| Yes | 16 | (19) | 1 | (3) | 5 | (12) | |
| *Missing* | 2 | | | | 1 | | |
| Family visit | | | | | | | |
| *Receive visitors >60 year of ag* | | | | | | | 0.99 |
| No | 75 | (89) | 30 | (91) | 36 | (88) | |
| Yes | 8 | (9) | 3 | (9) | 4 | (10) | |
| *Missing* | 2 | | | | 1 | | |
| *Visit people > 60 year of age* | | | | | | | 0.94 |
| No | 70 | (84) | 27 | (82) | 33 | (83) | |
| Yes | 13 | (16) | 6 | (18) | 7 | (17) | |
| *Missing* | 2 | | | | 1 | | |

(*Continued*)

**Table 2.** (Continued)

| Behaviour during COVID-19 crisis | N (%) | | N (%) | | N (%) | | P-value |
| --- | --- | --- | --- | --- | --- | --- | --- |
| | Negative-tested | | Positive-tested | | Control | | |
| Group gathering | | | | | | | 0.30 |
| No | 66 | (79) | 26 | (79) | 27 | (67) | |
| Yes, max 2 | 14 | (17) | 3 | (9) | 9 | (23) | |
| Yes 3–5 | 3 | (4) | 3 | (9) | 3 | (7) | |
| Yes 6–10 | 0 | (0) | 1 | (3) | 0 | (0) | |
| >10 | 0 | (0) | 0 | (0) | 1 | (3) | |
| *Missing* | *2* | | | | *1* | | |

Group comparison for non-pharmaceutical interventions (NPIs) and SARS-Cov-2 infection in the positive-tested, negative-tested (COVID-19 suspected), and control group (COVID-19 not suspected) analyses by the Kruskal-Wallis test.

(95%), however the control group reported adhering less to this 1.5- meter measure (p-value 0.02). There was no significant difference between the positive-tested, negative-tested, and control group in terms of other abovementioned risk behaviour (Table 2).

In this total study population, 72% reported always/almost always staying home during the COVID-19 pandemic; however, the control group reported staying home significantly less (p-value < 0.001). Thirty percent of all patients reported visiting shops (beyond the basic necessities of life) and 14% reported visiting markets, parks or beaches. The positive-tested group reported visiting markets, parks or beaches less often than the negative-tested group and control group (p-value 0.07). Approximately 16% and 9% of the study population reported visiting people older than 60 years and/or receiving visits from people aged 60 year or older, respectively. Attending group gatherings (> 6 people) was reported by 1% of the total study population.

## Mental state

The mean anxiety score was 48.56 points (range 23–67) in the negative-tested group and 50.10 points (range 27–77) in the positive-tested group (not significantly different). These anxiety scores of these COVID-19 suspected patients (mean 49.01, range 23.33–76.67) were significantly higher (p-value < 0.001) than the anxiety score of the control group (mean 39.00, range 20–70).

The anxiety scores of all patients were significantly inversely correlated to risk behaviour in the hygiene domain (ß 0.20, CI 0.93–7.62, p-value 0.01). The domains human contact and distancing were not significantly related to the anxiety scores in study group overall, with p-values of 0.87 and 0.89, respectively. The anxiety scores of the patients without comorbidities (mean 41.25, range 20–70) were significantly lower (ß 0.28, CI 3.47–11.79, p-value < 0.001) than the patients with comorbidities (48.88, range 23–77). Patients with a history of respiratory disease had significantly higher (ß 0.23, CI 1.83–10.42, p-value 0.05) anxiety scores (corrected for other comorbidities) than those without comorbidities.

## Discussion

The primary aim of this study was to investigate patient characteristics, knowledge of contamination risks, the severity of the disease, illness perception, and mental state in patients visiting an emergency department in the Netherlands during the COVID-19 pandemic. The

"COVID-19 suspected" (positive and negative tested group) was compared with the "COVID-19 not suspected" (control group) and in the "COVID-19 suspected" group, the positive and negative tested patients were compared.

The mean STAI-s score, which represents anxiety, was significantly higher in the COVID-19 suspected patients compared to the general population requiring emergency care. No significant difference in the anxiety score was observed in the positive-tested and negative-tested patients. Moreover, COVID-19 suspected patients and patients with (pulmonary) comorbidities were significantly more anxious, and the higher the degree of fear, the more seriously the hygiene measures were followed.

Remarkably, more than 50% of the positive-tested patients worked in health care and most of them in the medical sector. The percentage of the SARS-Cov-2 positive-tested group who reported having visited public places was lower than in the other two groups. The control group reported adhering less to the 1.5- meter measure and stayed at home less.

Furthermore, the severity of SARS-Cov-2 infection, the risk of contamination and the importance of NPIs were well understood by all patients, and the measures were therefore properly observed in the positive-tested, negative-tested, and control group. The age of the patients in this study was quite young (mean age of 50.42 years).

Significantly higher anxiety was observed in the COVID-19 suspected patients than reported in the control group and the average research population [5]. This is in line with the large degree of fear that was previously reported by Liu et al., stating that patients with COVID-19 experienced high levels of anxiety (mean STAI score of 58) and low sleep quality [6]. The current study adds to these findings that this anxiety is already present in suspected patients and not only in diagnosed patients or in the total population. There was thus no mass hysteria regarding COVID-19. In line with our study, Motta Zanin et al. investigated the public Italian perception of health risk through the administration of a questionnaire in more and less COVID-19 effected regions. They demonstrated that the participants mainly expressed uncertainty, fear, and sadness in the more effected regions [7]. Mainly, patients with (pulmonary) comorbidities were significantly more anxious in our study, and the higher the fear, the better hygiene measures were followed.

The percentage of SARS-Cov-2 infected health care professionals in our study is higher than that reported by Heinzerling et al. [8]. In that study involving 121 hospital health care workers who were exposed to COVID-19 patients in a Hospital in California, 36% developed symptoms during 14 days after exposure and only 3 people tested positive. Another study demonstrated that 92% of the health care personnel in the United States had at least one symptom (fever, cough, or shortness of breath) after exposure to SARS-Cov-2-infected patients, however this was not proven to be COVID-19-related [9]. Barrett et al. demonstrated that the prevalence of SARS-CoV-2 infection was higher among health care workers (7.3%) than in non-health care workers (0.4%) [10].

The severity of SARS-Cov-2 infection, the risk of contamination and the importance of NPIs were well understood by all patients and the measures to prevent infection were (therefore) reported as being adhered to in this study. In line with our study, a cross-sectional survey in Hong Kong with 765 participants demonstrated that the overall knowledge and understanding of COVID-19 was good and most respondents agreed that NPIs could reduce the transmission of COVID-19 [11].

There was clear communication from the Dutch government about the importance of NPIs to prevent transmission of SARS-Cov-2. The study of Gesser-Edelsburg et al. underscores this importance of the public trust in adhering to NPIs to prevent transmission of SARS-Cov-2. They showed in an online survey in the Israeli public with 1056 participants that the higher the

public trust and evaluation of crisis management was, the greater the compliance of the public with the government guidelines [12].

The percentage who reported using a face mask in this study was much lower compared to a study in China, which found 97% wore face masks [13]. At the beginning of the COVID-19 period, face masks were not compulsory and often even not available in nursing homes and care institutions in the Netherlands.

The lower average age in this study compared to the SARS-Cov-2-infected populations reported in other studies can be explained by the selection criteria for referring patients to the hospital in the Netherlands. General practitioners refer patients with relatively severe symptoms and patients for whom health improvements could reasonably be expected, per the Dutch referral policy. This policy excludes patients with mild symptoms and vulnerable elderly patients not eligible for treatment in the hospital. General practitioners and nursing home doctors take into account whether medical treatment is desirable and useful in respect to the quality of life of each individual patient. This approach is in contrast to a retrospective cohort study included 124 patients who required 911 Emergency Medical Services care for COVID-19 in Washington. This study revealed that most patients with COVID-19 presenting to emergency medical services were older and had multiple chronic health conditions [14].

This is one of the first (large) study that investigates and compares patient characteristics, knowledge, behaviour, illness perception, and mental state with respect to COVID-19 of patients visiting the emergency room, subdivided as being suspected of having COVID-19 (who then tested either positive or negative) and a control group not suspected of having COVID-19. The limitations of this study include the state of COVID-19 testing and measures taken by the population in the Netherlands, which may not be generalizable for other countries. Also, seriously ill patients who were respiratory insufficient, patients with a language barrier, and patients who were not able to handle an iPad were excluded from this study. Finally, the data on risk behaviours were self-reported, which implies that social desirability could have played a role in the low risk behaviours described in this sample.

Regulatory agencies urge adherence to NPIs to reduce SARS-Cov-2 virus contamination [15, 16]. In the group suspected of infection, risk behaviour was no greater in the group that tested positive than in the group that tested negative. High exposure to SARS-Cov-2 in the health care sector was probably the most important factor in being infected with COVID-19. This specific patient group should try to prevent infection by using protective material, including face masks. However, at the beginning of the COVID-19 crisis most of the available protective equipment was destined for hospitals and there was a shortage in nursing homes and care institutions. This may explain the high number of infections in nursing homes and care institutions.

However, the disproportionately large number of infections in health care workers may also suggest a super spreading pattern [17] as observed in church choirs, gyms, and animal slaughterhouses. Health care professionals often work and stay in poorly ventilated rooms, as do workers in nursing homes and care institutions. Furthermore, aerosol generating procedures (tracheal intubation and non-invasive ventilation) increase the risk of transmission of SARS-Cov-2 to health care workers [18]. The health care workers in this study mainly worked in health care institutions not involved in these procedures.

## Conclusion

This is the first (large) study that investigates and compares patient characteristics, knowledge, behaviour, illness perception, and mental state with respect to COVID-19 of patients visiting the emergency room, subdivided as being suspected of having COVID-19 and a control group

not suspected of having COVID-19. All patients in this study were generally aware of transmission risks and virulence and adhered to the NPIs. Most patients suspected of having COVID-19 who visited our emergency department had a high degree of anxiety compared to the general population. There is thus no mass hysteria regarding COVID-19, but anxiety was higher among the patients suspected of having COVID-19. Patients with (pulmonary) comorbidities were significantly more anxious and experienced a higher degree of fear, which was correlated with better behaviour as regards hygiene measures. Knowledge about the coping of the population during the COVID-19 pandemic is very important, certainly also in the perspective of a possible second outbreak of COVID-19.

## Evidence before this study

We searched PubMed using the keywords "COVID-19", "2019-nCoV", "SARS-CoV-2" and "emergency", "anxiety", "mental state", "fear", "patient characteristics", "knowledge", "risk behaviour". We screened 90 articles (updated 21th of August 2020) and considered 4 studies relevant to our research goal. We also checked references to relevant articles ('snowballing').

*(COVID-19[Title] OR 2019-nCoV[Title] OR SARS-CoV-2[Title]) AND emergency[Title] AND (anxiety OR mental state OR fear) AND (patient characteristics OR knowledge OR risk behaviour).*

## Added value of this study

We investigate patient characteristics, knowledge of contamination risks, the severity of the disease, illness perception, and mental state in patients visiting an emergency department in the Netherlands during the COVID-19 pandemic. We found that the severity of SARS-Cov-2 infection, the risk of contamination and the importance of NPIs were well understood. Most patients suspected of having COVID-19 had a high degree of anxiety compared to the general population. Patients with (pulmonary) comorbidities were significantly more anxious which was correlated with better behaviour as regards hygiene measures. There is thus no mass hysteria regarding COVID-19, but anxiety was higher among the patients suspected of having COVID-19.

## Implications of all the available evidence

Clear communication from the government about the importance of NPIs to prevent transmission of SARS-Cov-2 ensures the adherence to NPIs. High degree of anxiety is present in COVID-19 suspected patients and in the patients with (pulmonary) comorbidities. It is important to pay attention to this psychological aspect of COVID-19 in this particular part of the population. Knowledge about the coping of the population during the COVID-19 pandemic is very important, certainly also in the perspective of a possible second outbreak of COVID-19.

## Patient and public involvement

No patient involved.

## Supporting information

**S1 Questionnaire. Questionnaire on knowledge and illness perception.**
(DOCX)

**S2 Questionnaire. Questionnaire on risk behaviour.**
(DOCX)

**S1 Protocol.**
(PDF)

**S2 Protocol.**
(PDF)

**S3 Protocol.**
(DOCX)

**S1 Data.**
(XLSX)

**S2 Data.**
(XLSX)

**S1 File.**
(PDF)

**S2 File.**
(JPG)

**S3 File.**
(JPG)

## Author Contributions

**Data curation:** J. P. M. van der Valk, F. W. J. Heijboer.

**Formal analysis:** J. P. M. van der Valk, F. W. J. Heijboer.

**Methodology:** J. P. M. van der Valk, H. van Middendorp.

**Supervision:** J. C. C. M. in 't Veen.

**Writing – original draft:** J. P. M. van der Valk.

**Writing – review & editing:** H. van Middendorp, A. W. M. Evers, J. C. C. M. in 't Veen.

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
