## [Decision Letter · Decision Letter 0]

18 Jan 2021

PONE-D-20-28005

Case- control study of patient characteristics, knowledge of the COVID-19 disease, risk behaviour and mental state in patients visiting an emergency room with COVID-19 symptoms in the Netherlands

PLOS ONE

Dear Dr. Johanna van der Valk,

Thank you for submitting your manuscript to PLOS ONE. After careful consideration, we feel that it has merit but does not fully meet PLOS ONE’s publication criteria as it currently stands. Therefore, we invite you to submit a revised version of the manuscript that addresses the points raised during the review process.

We look forward to receiving your revised manuscript.

Kind regards,

Yuka Kotozaki

Academic Editor

PLOS ONE

Journal Requirements:

2. Please include additional information regarding the survey or questionnaire used in the study and ensure that you have provided sufficient details that others could replicate the analyses. For instance, if you developed a questionnaire as part of this study and it is not under a copyright more restrictive than CC-BY, please include a copy, in both the original language and English, as Supporting Information.  If the original language is written in non-Latin characters, for example Amharic, Chinese, or Korean, please use a file format that ensures these characters are visible.

3. Please state whether you validated the questionnaire prior to testing on study participants. Please provide details regarding the validation group within the methods section.

4. For more information on PLOS ONE's expectations for statistical reporting, please see https://journals.plos.org/plosone/s/submission-guidelines.#loc-statistical-reporting. Please update your Methods and Results sections accordingly.

Reviewers' comments:

Reviewer's Responses to Questions

**Comments to the Author**

1. Is the manuscript technically sound, and do the data support the conclusions?

Reviewer #1: Partly

Reviewer #2: No

2. Has the statistical analysis been performed appropriately and rigorously? 

Reviewer #1: I Don't Know

Reviewer #2: No

3. Have the authors made all data underlying the findings in their manuscript fully available?

Reviewer #1: Yes

Reviewer #2: No

4. Is the manuscript presented in an intelligible fashion and written in standard English?

Reviewer #1: Yes

Reviewer #2: Yes

5. Review Comments to the Author

Reviewer #1: Important note: This review pertains only to ‘statistical aspects’ of the study and so ‘clinical aspects’ [like medical importance, relevance of the study, ‘clinical significance and implication(s)’ of the whole study, etc.] are to be evaluated [should be assessed] separately/independently. Further please note that any ‘statistical review’ is generally done under the assumption that (such) study specific methodological [as well as execution] issues are perfectly taken care of by the investigator(s). This review is not an exception to that and so does not cover clinical aspects {however, seldom comments are made only if those issues are intimately / scientifically related & intermingle with ‘statistical aspects’ of the study}. Agreed that ‘statistical methods’ are used as just tools here, however, they are vital part of methodology [and so should be given due importance].

COMMENTS: It is (being KAP type cross-sectional survey only) a fairly simple [and so straight forward] study as ‘The primary aim of this study was to investigate “Patient characteristics, knowledge of the COVID-19 disease, risk behaviour and illness perception in patients visiting an emergency department in the Netherlands during the COVID-19 pandemic”. However, as said in (lines 63-64) ‘Abstract-Conclusion’ that “This is the first (large) study that investigates these’ is not true. But in any case, I request the authors to consider/note following points:

It may please be noted {kindly confirm from field experts} that patients having SARS-Cov-2 antibodies are not capable to spread the disease [unless IgM result shows status of infection]. If that is so, then how correct is to combine them? with ‘Patients with a positive nasal Polymerase chain reaction (PCR) swab to SARS-Cov-2’ which are real ‘COVID-19 positive’.

Though measures/tools used as “Indicators/Measures of knowledge, risk behaviour, and illness perception of COVID-19” (lines 155-56), are appropriate, most of them yield data that are in [most likely] ‘ordinal’ level of measurement [and not in ratio level of measurement for sure {as the score two times higher does not indicate presence of that parameter/phenomenon as double}]. Then application of suitable non-parametric test(s) is/are indicated/advisable [even if distribution may be ‘Gaussian’ (i.e. normal) in these (such) cases. Therefore, as said in lines 157-8 that ‘differences in terms of these factors were compared between 3 groups (those tested positive for SARS-Cov-2, those tested negative, and the control group) using the One-Way ANOVA test’ is not correct and it is indicated/advisable to use non-parametric ‘One-Way ANOVA’ namely Kruskal-Wallis test.

Please read the following [from famous text book]:

“Inferential statistics (i.e. hypothesis testing + estimation of CI) is built on the population model (i.e. the underlying assumption is that there is a population and we are dealing with random sample(s) drawn from that population). Although in clinical trial (involving at least two groups) we do not really deal with random samples, ‘allocation’ to treatment groups is ‘randomly’ done which enable us to evoke the population model and we can use inferential statistics safely. But when there is only one group or in studies even with two/more groups ‘random allocation’ is out-of- question [like internal grouping as in this case] and with ‘non-random’ selection, it may be questionable to use inferential statistics [even if you have two measurement sets as ‘pre-post’].

By this I do not advice “not to use inferential statistics here”, but just to keep this limitation in mind while interpreting results as there is no guarantee of representation of population {example, line 336: The lower average age in this study compared to the SARS-Cov-2-infected populations reported in other studies}.

Test used to analyse data displayed in Table 2 [Group comparison for non-pharmaceutical interventions (NPIs) and SARS-Cov-2 infection in the positive-tested, negative-tested, and control groups] is not mentioned anywhere and so the question is ‘how ANOVA is applied as most data are categorical’ but if Chi-square is used then ‘how zero frequency (rather all low frequencies are/) is dealt with’ [remember that this a scientific/academic document and so all details should be clearly communicated].

Implications of this study [in backdrop of ‘Added value of this study’ described in lines 396-40] are questionable, in my opinion {though it is true (line 71-72) that ‘Knowledge about the coping of the population during the COVID-19 pandemic is very important, certainly also in the perspective of a possible second outbreak of COVID-19’}. Few things/findings are ‘very obvious’ [ex. Line 186-7: ‘Significantly more patients infected with SARS-Cov-2 work in a vital profession compared to uninfected patients and the control group (p-value 0.04), and most of them work in the medical sector’].

Reviewer #2: This is an ambitious case- control study of patient characteristics, knowledge of the COVID-19 disease, risk behaviour and mental state in patients visiting an emergency room with COVID-19 symptoms and controls in the Netherlands. I admire the intent and work to perform this work, but unfortunately the study population, questionnaire with responses, and analyses require clarification. The study population as it is described decreases the study generalizability. Measuring behaviour by self report in patients with COVID-19 symptoms compared to those without COVID-19 symptoms might be too confounding to say anything about behaviour. Use of the anxiety measure without reference or definition of the meaning of the scoring system decreases its utility. Use of a questionnaire without validation or precision is worrisome. (For example when asked about staying home is that staying home aside from work or inclusive, is it in general (which is what I thought initially) or with symptoms or post exposure (which is what the questionnaire seems to imply ? Analysis of the questionnaire results with binary outcomes when they were measured on a 5 point likert scale is puzzling and very much decreases belief or precision in the results. For example do the essential worker patients, such as those who work in a nursing home really keep a 1.5m distance if they are working with patients? Is going to a park for excersie or to walk the dog an exposure? But similarly the only intervention reported on a 5 point scale was handwashing which I would imagine is very hard to recall correctly.

1) Study population: Consecutive patients presenting at one ED in the Netherlands between April 10 and June 20. Because this is one ED it is important to understand the ED and patient population in order to see the extent to which this study is generalizable. 159 patients were enrolled in approximately 70 days. This is 2.5 patients a day which seems like a very low volume ED or it is a convenience enrollment. In either case, it is important to understand if there was a difference between those that enrolled and those that did not enroll in the study. It would be good to understand the chief complaints/diagnostic categories of the cases and controls as well as their disposition and illness severity. I want to know what are the parameters for testing “covid suspected.” Would it be symptomatic patients or exposed? If so, there are some constellations of symptoms that are more worrisome than others (fever alone, vs. respiratory symptoms and fever.) What was the prevalence in the netherlads at that time?

2) Comparison is really between covid suspected and controls then a subanalysis could be done with positive and negatives.

3) Since this study involves patient recall regarding behavior, it is important to use validated questions /outcome measurements regarding behavior. It is important to understand the Netherlands rules/public health injunctions at this time (Were masks required?) The authors report using a 5 point likert scale but the report their results on a 2 point scale which seems a little blunt

4) Recall bias possible regarding covid suspected and covid positive (were they feeling sicker)

5) Since this is recall, how long of a time period are they asking the patient to recall?

6) Likewise, visiting beaches, markets and parks seems vague. I do not think of beaches and parks as places with increased transmission (in the US we are allowed to go to parks and beaches and out of doors.)

7) If the authors are going to use an anxiety score they should cite it and then describe its use. I had to look it up on the internet. While the authors state that the comparison of anxiety measurements are significantly different, I do not know what this means. My source says that the STAI-s is used to make clinical diagnoses but I don’t think that is the case here. The source I saw on the internet stated that the ranges found here are both “moderate”

6. PLOS authors have the option to publish the peer review history of their article (what does this mean?). If published, this will include your full peer review and any attached files.

Reviewer #1: No

Reviewer #2: No

---

## [Author Response · Author response to Decision Letter 0]

26 Jan 2021

Dear Editor

Thank you very much for giving us the opportunity to revise our article titled: Case- control study of patient characteristics, knowledge of the COVID-19 disease, risk behaviour and mental state in patients visiting an emergency room with COVID-19 symptoms in the Netherlands”

We are very thankful for the commentary of the referees. We would like to respond point by point to the comments. For your convenience, we have written the answers in blue.

COMMENTS – Manuscript PONE-D-20-28005

Title: “Case- control study of patient characteristics, knowledge of the COVID-19 disease, risk behaviour and mental state in patients visiting an emergency room with COVID-19 symptoms in the Netherlands”

Reviewer #1.COMMENTS: It is (being KAP type cross-sectional survey only) a fairly simple [and so straight forward] study as ‘The primary aim of this study was to investigate “Patient characteristics, knowledge of the COVID-19 disease, risk behaviour and illness perception in patients visiting an emergency department in the Netherlands during the COVID-19 pandemic”. However, as said in (lines 63-64) ‘Abstract-Conclusion’ that “This is the first (large) study that investigates these’ is not true. 

Thank you for this comment, we agree with you. We changed the sentence to: 

‘this is one of the first (large) study that investigates …’

But in any case, I request the authors to consider/note following points:

It may please be noted {kindly confirm from field experts} that patients having SARS-Cov-2 antibodies are not capable to spread the disease [unless IgM result shows status of infection]. If that is so, then how correct is to combine them? with ‘Patients with a positive nasal Polymerase chain reaction (PCR) swab to SARS-Cov-2’ which are real ‘COVID-19 positive’.

We agree with you. However, only 3 patients were included only on the SARS-Cov-2 antibodies. All other patients were tested with COVID-19 PCR. Because, this study was performed in the beginning of the pandemic, we assume that these 3 patients had a very recent infection. We think that taking these 3 patients in to account will not influence our study results. 

Though measures/tools used as “Indicators/Measures of knowledge, risk behaviour, and illness perception of COVID-19” (lines 155-56), are appropriate, most of them yield data that are in [most likely] ‘ordinal’ level of measurement [and not in ratio level of measurement for sure {as the score two times higher does not indicate presence of that parameter/phenomenon as double}]. Then application of suitable non-parametric test(s) is/are indicated/advisable [even if distribution may be ‘Gaussian’ (i.e. normal) in these (such) cases. Therefore, as said in lines 157-8 that ‘differences in terms of these factors were compared between 3 groups (those tested positive for SARS-Cov-2, those tested negative, and the control group) using the One-Way ANOVA test’ is not correct and it is indicated/advisable to use non-parametric ‘One-Way ANOVA’ namely Kruskal-Wallis test.

Thank you for this good comment. We agree with you and did the analysis again with the non-parametric One-Way ANOVA test, namely Kruskal-Wallis test. We have adjusted table 2. Fortunately, the new results of this statistical method did not change our conclusions. 

Please read the following [from famous text book]:

“Inferential statistics (i.e. hypothesis testing + estimation of CI) is built on the population model (i.e. the underlying assumption is that there is a population and we are dealing with random sample(s) drawn from that population). Although in clinical trial (involving at least two groups) we do not really deal with random samples, ‘allocation’ to treatment groups is ‘randomly’ done which enable us to evoke the population model and we can use inferential statistics safely. But when there is only one group or in studies even with two/more groups ‘random allocation’ is out-of- question [like internal grouping as in this case] and with ‘non-random’ selection, it may be questionable to use inferential statistics [even if you have two measurement sets as ‘pre-post’].

By this I do not advice “not to use inferential statistics here”, but just to keep this limitation in mind while interpreting results as there is no guarantee of representation of population {example, line 336: The lower average age in this study compared to the SARS-Cov-2-infected populations reported in other studies}.

Test used to analyse data displayed in Table 2 [Group comparison for non-pharmaceutical interventions (NPIs) and SARS-Cov-2 infection in the positive-tested, negative-tested, and control groups] is not mentioned anywhere and so the question is ‘how ANOVA is applied as most data are categorical’ but if Chi-square is used then ‘how zero frequency (rather all low frequencies are/) is dealt with’ [remember that this a scientific/academic document and so all details should be clearly communicated].

Thank you very much for the book suggestion. We have read the book chapter with interest. We used indeed the ANOVA test and as you mentioned, it is better to use the Kruskal-Wallis test. We made this improvement and correct this in the statistical methods. We have also adapted table 2 and noted the statistical method below the table. 

Implications of this study [in backdrop of ‘Added value of this study’ described in lines 396-40] are questionable, in my opinion {though it is true (line 71-72) that ‘Knowledge about the coping of the population during the COVID-19 pandemic is very important, certainly also in the perspective of a possible second outbreak of COVID-19’}. Few things/findings are ‘very obvious’ [ex. Line 186-7: ‘Significantly more patients infected with SARS-Cov-2 work in a vital profession compared to uninfected patients and the control group (p-value 0.04), and most of them work in the medical sector’].

Maybe, these conclusions of our study are obvious. However, we think it is important to pay attention to this important sector and therefore, we mentioned this point in our manuscript. 

Reviewer #2: This is an ambitious case- control study of patient characteristics, knowledge of the COVID-19 disease, risk behaviour and mental state in patients visiting an emergency room with COVID-19 symptoms and controls in the Netherlands. I admire the intent and work to perform this work, but unfortunately the study population, questionnaire with responses, and analyses require clarification. The study population as it is described decreases the study generalizability. Measuring behaviour by self report in patients with COVID-19 symptoms compared to those without COVID-19 symptoms might be too confounding to say anything about behaviour. 

Thank you very much for this extensive comment. The questionnaires were completed before the patients get there COVID-19 PCR result, therefore, there is no confounding in the questionnaire outcomes.

Use of the anxiety measure without reference or definition of the meaning of the scoring system decreases its utility. 

We agree with you and added the reference to the manuscript: Marteau TM, Bekker H. The development of a six-item short-form of the state scale of the Spielberger State-Trait Anxiety Inventory (STAI). Br J Clin Psychol. 1992;31(3):301-6.

A cut off point of 39–40 has been suggested to detect clinically significant anxiety symptoms and we clarified this in the manuscript. 

Use of a questionnaire without validation or precision is worrisome. (For example when asked about staying home is that staying home aside from work or inclusive, is it in general (which is what I thought initially) or with symptoms or post exposure (which is what the questionnaire seems to imply ? Analysis of the questionnaire results with binary outcomes when they were measured on a 5 point likert scale is puzzling and very much decreases belief or precision in the results. For example do the essential worker patients, such as those who work in a nursing home really keep a 1.5m distance if they are working with patients? Is going to a park for excersie or to walk the dog an exposure? But similarly the only intervention reported on a 5 point scale was handwashing which I would imagine is very hard to recall correctly.

That is a very good comment. At the start of the study we were looking for validated questionnaires about behaviour during infection outbreaks. Unfortunately, validated questionaries don’t exist on this topic. And it is true that these questionnaires have a recall bias. However, the time between the start of the pandemic and completing the interview is only 3.5 month. 

Study population: Consecutive patients presenting at one ED in the Netherlands between April 10 and June 20. Because this is one ED it is important to understand the ED and patient population in order to see the extent to which this study is generalizable. 159 patients were enrolled in approximately 70 days. This is 2.5 patients a day which seems like a very low volume ED or it is a convenience enrollment. In either case, it is important to understand if there was a difference between those that enrolled and those that did not enroll in the study. It would be good to understand the chief complaints/diagnostic categories of the cases and controls as well as their disposition and illness severity. I want to know what are the parameters for testing “covid suspected.” Would it be symptomatic patients or exposed? 

All patients with upper- or lower respiratory symptoms were considered as possible COVID-positive and were tested with PCR. We added the following sentence to the manuscript:

All patients with upper- or lower respiratory symptoms were considered as possible COVID-19 positive.

If so, there are some constellations of symptoms that are more worrisome than others (fever alone, vs. respiratory symptoms and fever.) What was the prevalence in the netherlads at that time?

The prevalence of COVID-19 suspected patients in our emergency room was among 7 patients a day. There is indeed a selection bias, because the seriously ill patients who were respiratory insufficient, patients with a language barrier, and patients who were not able to handle an iPad were excluded from this study. Unfortunately, that was inevitable. We did mention this in the manuscript. 

2) Comparison is really between covid suspected and controls then a subanalysis could be done with positive and negatives.

We have chosen to compare the positive, negative and not suspected COVID-19 positive patients to get a more complete vision of the metal state and behaviour of all patients visiting the emergency room. However, it is possible to get the comparison between the positive and negative patients from the tables.

3) Since this study involves patient recall regarding behavior, it is important to use validated questions /outcome measurements regarding behavior. It is important to understand the Netherlands rules/public health injunctions at this time (Were masks required?) The authors report using a 5 point likert scale but the report their results on a 2 point scale which seems a little blunt

We choose to report not all details to make it easy for the reader. We think is a more organized and less extensive table. However, it is indeed also possible to give the total overview of all numbers,

4) Recall bias possible regarding covid suspected and covid positive (were they feeling sicker)

That is a good point. That maybe the case. Hoverer, we have not scored the degree of illness in this study. 

5) Since this is recall, how long of a time period are they asking the patient to recall?

The recall was maximal 3.5 month. 

6) Likewise, visiting beaches, markets and parks seems vague. I do not think of beaches and parks as places with increased transmission (in the US we are allowed to go to parks and beaches and out of doors.)

In the Netherlands it is very busy on the beaches and parks. Therefore, these places are considered as places with a high risk of infection.

7) If the authors are going to use an anxiety score they should cite it and then describe its use. I had to look it up on the internet. While the authors state that the comparison of anxiety measurements are significantly different, I do not know what this means. My source says that the STAI-s is used to make clinical diagnoses but I don’t think that is the case here. The source I saw on the internet stated that the ranges found here are both “moderate”

We agree with you. We added a reference to the manuscript. 

The mean anxiety score was 48.56 points (range 23-67) in the negative-tested group and 50.10 points (range 27-77) in the positive-tested group. The the anxiety score of the control group (mean 39.00, range 20-70). As we mentioned in the manuscript a cut off point of 39–40 has been suggested to detect clinically significant anxiety symptoms. We have clarified this in the text. 

We hope we have answered all questions and really hope you will accept our improved manuscript in Plos One. I would like to ask you to take into consideration that it will be very difficult for us to get our article published in another journal after a 4.5 month waiting period for review at Plos one. 

Looking forward to hear from you. 

Kind regards,

Hanna van der Valk

---

## [Decision Letter · Decision Letter 1]

3 Mar 2021

PONE-D-20-28005R1

Case- control study of patient characteristics, knowledge of the COVID-19 disease, risk behaviour and mental state in patients visiting an emergency room with COVID-19 symptoms in the Netherlands

PLOS ONE

Dear Dr. Johanna van der Valk,

Thank you for submitting your manuscript to PLOS ONE. After careful consideration, we feel that it has merit but does not fully meet PLOS ONE’s publication criteria as it currently stands. Therefore, we invite you to submit a revised version of the manuscript that addresses the points raised during the review process.

We look forward to receiving your revised manuscript.

Kind regards,

Yuka Kotozaki

Academic Editor

PLOS ONE

Reviewers' comments:

Reviewer's Responses to Questions

**Comments to the Author**

1. If the authors have adequately addressed your comments raised in a previous round of review and you feel that this manuscript is now acceptable for publication, you may indicate that here to bypass the “Comments to the Author” section, enter your conflict of interest statement in the “Confidential to Editor” section, and submit your "Accept" recommendation.

Reviewer #1: (No Response)

Reviewer #2: (No Response)

2. Is the manuscript technically sound, and do the data support the conclusions?

Reviewer #1: Yes

Reviewer #2: Partly

3. Has the statistical analysis been performed appropriately and rigorously? 

Reviewer #1: N/A

Reviewer #2: I Don't Know

4. Have the authors made all data underlying the findings in their manuscript fully available?

Reviewer #1: Yes

Reviewer #2: Yes

5. Is the manuscript presented in an intelligible fashion and written in standard English?

Reviewer #1: Yes

Reviewer #2: Yes

6. Review Comments to the Author

Reviewer #1: COMMENTS: Since most of the comments made on earlier draft by me (and hopefully by other respected reviewers also) are attended positively/adequately, I am fully satisfied and the manuscript is improved a lot.

While answering my comment, it is said that “Fortunately, the new results of this statistical method did not change our conclusions” is very good but remember that it does not indicate that any will do. Methodology used should be appropriate [always use which is indicated / most desired].

Reviewer #2: (No Response)

7. PLOS authors have the option to publish the peer review history of their article (what does this mean?). If published, this will include your full peer review and any attached files.

Reviewer #1: **Yes: **Dr. Sanjeev Sarmukaddam

Reviewer #2: No

---

## [Author Response · Author response to Decision Letter 1]

4 Mar 2021

Dear Editor

We would like to respond to the comment of the referee.

1. "While answering my comment, it is said that “Fortunately, the new results of this statistical method did not change our conclusions” is very good but remember that it does not indicate that any will do. Methodology used should be appropriate [always use which is indicated / most desired."

I fully agree with you. I am very pleased that you pointed this out to me, and indeed the correct methodology is very important. We want to apologize for the word choice we made. 

Looking forward to hear from you. 

Kind regards,

Hanna van der Valk

---

## [Decision Letter · Decision Letter 2]

15 Mar 2021

PONE-D-20-28005R2

Case- control study of patient characteristics, knowledge of the COVID-19 disease, risk behaviour and mental state in patients visiting an emergency room with COVID-19 symptoms in the Netherlands

PLOS ONE

Dear Dr. Johanna van der Valk,

Thank you for submitting your manuscript to PLOS ONE. After careful consideration, we feel that it has merit but does not fully meet PLOS ONE’s publication criteria as it currently stands. Therefore, we invite you to submit a revised version of the manuscript that addresses the points raised during the review process.

We look forward to receiving your revised manuscript.

Kind regards,

Yuka Kotozaki

Academic Editor

PLOS ONE

Journal Requirements:

Reviewers' comments:

Reviewer's Responses to Questions

**Comments to the Author**

1. If the authors have adequately addressed your comments raised in a previous round of review and you feel that this manuscript is now acceptable for publication, you may indicate that here to bypass the “Comments to the Author” section, enter your conflict of interest statement in the “Confidential to Editor” section, and submit your "Accept" recommendation.

Reviewer #1: (No Response)

2. Is the manuscript technically sound, and do the data support the conclusions?

Reviewer #1: (No Response)

3. Has the statistical analysis been performed appropriately and rigorously? 

Reviewer #1: (No Response)

4. Have the authors made all data underlying the findings in their manuscript fully available?

Reviewer #1: (No Response)

5. Is the manuscript presented in an intelligible fashion and written in standard English?

Reviewer #1: (No Response)

6. Review Comments to the Author

Reviewer #1: COMMENTS: As already said, it being a KAP type cross-sectional survey only [is fairly simple and so straight forward], there is hardly anything to comment, mention/point-out critical observation(s) or suggest few things for further improvement.

However, I just felt that the fact ‘this study also includes comparison between three groups [Negative-tested, Positive-tested, Control] is not adequately mentioned [except line 64 of ‘Abstract-Conclusion’ that “This is one of the first (large) study that investigates and compares patient characteristics”]

7. PLOS authors have the option to publish the peer review history of their article (what does this mean?). If published, this will include your full peer review and any attached files.

Reviewer #1: **Yes: **Dr. Sanjeev Sarmukaddam

---

## [Author Response · Author response to Decision Letter 2]

19 Mar 2021

Dear Editor

We would like to respond to the comment of the referee.

Reviewer #1: COMMENTS: As already said, it being a KAP type cross-sectional survey only [is fairly simple and so straight forward], there is hardly anything to comment, mention/point-out critical observation(s) or suggest few things for further improvement.

However, I just felt that the fact ‘this study also includes comparison between three groups [Negative-tested, Positive-tested, Control] is not adequately mentioned [except line 64 of ‘Abstract-Conclusion’ that “This is one of the first (large) study that investigates and compares patient characteristics”]

We agree with you. As you suggest, we mentioned now more adequately in our manuscript the fact that this study describes the variables in patients visiting an emergency department in the Netherlands during the COVID-19 pandemic and that we also made a comparison between the “COVID-19 suspected” (positive and negative tested group) with the “COVID-19 not suspected” (control group) and in the “COVID-19 suspected” group, the positive and negative tested patients. 

We added the following sentences to the manuscript:

Abstract (line 46-49)

The primary aim of this study was to investigate these variables in patients visiting an emergency department in the Netherlands during the COVID-19 pandemic and to compare the “COVID-19 suspected” (positive and negative tested group) with the “COVID-19 not suspected” (control group) and to compare in the “COVID-19 suspected” group, the positive and negative tested patients.

Material and methods (line 160-162)

A comparison is made between the “COVID-19 suspected” (positive and negative tested group) with the “COVID-19 not suspected” (control group) and in the “COVID-19 suspected” group, the positive and negative tested patients.

We clarified this also in the discussion (line 314-317)

The “COVID-19 suspected” (positive and negative tested group) was compared with the “COVID-19 not suspected” (control group) and in the “COVID-19 suspected” group, the positive and negative tested patients were compared.

We really hope -by addressing you comment- that we improved the message of the manuscript. We are looking forward to hear from you.

Kind regards,

Hanna van der Valk

---

## [Decision Letter · Decision Letter 3]

26 Mar 2021

Case- control study of patient characteristics, knowledge of the COVID-19 disease, risk behaviour and mental state in patients visiting an emergency room with COVID-19 symptoms in the Netherlands

PONE-D-20-28005R3

Dear Dr. Johanna van der Valk,

We’re pleased to inform you that your manuscript has been judged scientifically suitable for publication and will be formally accepted for publication once it meets all outstanding technical requirements.

Kind regards,

Yuka Kotozaki

Academic Editor

PLOS ONE

Additional Editor Comments (optional):

Reviewers' comments:

Reviewer's Responses to Questions

**Comments to the Author**

1. If the authors have adequately addressed your comments raised in a previous round of review and you feel that this manuscript is now acceptable for publication, you may indicate that here to bypass the “Comments to the Author” section, enter your conflict of interest statement in the “Confidential to Editor” section, and submit your "Accept" recommendation.

Reviewer #1: (No Response)

2. Is the manuscript technically sound, and do the data support the conclusions?

Reviewer #1: (No Response)

3. Has the statistical analysis been performed appropriately and rigorously? 

Reviewer #1: (No Response)

4. Have the authors made all data underlying the findings in their manuscript fully available?

Reviewer #1: (No Response)

5. Is the manuscript presented in an intelligible fashion and written in standard English?

Reviewer #1: (No Response)

6. Review Comments to the Author

Reviewer #1: COMMENTS: As already said, it being a KAP type cross-sectional survey is fairly simple and straight forward, therefore there is nothing much to comment as it describes factual information. I think if this info be useful [if at all] for clinicians involved in treating COVID patients, then we should not delay its publication.

Hope, you already have taken note of the fact {which was highlighted on very first occasion by including ‘important note’} that “This review pertains only to ‘statistical aspects’ of the study and so ‘clinical aspects’ [like medical importance, relevance of the study, ‘clinical significance and implication(s)’ of the whole study, etc.] are to be evaluated [should be assessed] separately/independently”.

7. PLOS authors have the option to publish the peer review history of their article (what does this mean?). If published, this will include your full peer review and any attached files.

Reviewer #1: **Yes: **Dr. Sanjeev Sarmukaddam

---

## [Editor Report · Acceptance letter]

6 Apr 2021

PONE-D-20-28005R3 

Case- control study of patient characteristics, knowledge of the COVID-19 disease, risk behaviour and mental state in patients visiting an emergency room with COVID-19 symptoms in the Netherlands 

Dear Dr. van der Valk:

I'm pleased to inform you that your manuscript has been deemed suitable for publication in PLOS ONE. Congratulations! Your manuscript is now with our production department. 

Kind regards, 

on behalf of

Dr. Yuka Kotozaki 

Academic Editor

PLOS ONE